# A Platform of Unmanned Surface Vehicle Swarms for Real Time Monitoring in Aquaculture Environments

**DOI:** 10.3390/s19214695

**Published:** 2019-10-29

**Authors:** Daniela Sousa, Diego Hernandez, Francisco Oliveira, Miguel Luís, Susana Sargento

**Affiliations:** 1Instituto de Telecomunicações, 3810-193 Aveiro, Portugal; dc.hernandez@ua.pt (D.H.); fbpo@ua.pt (F.O.); nmal@av.it.pt (M.L.); susana@ua.pt (S.S.); 2Departamento de Eletrónica, Telecomunicações e Informática, University of Aveiro, 3810-193 Aveiro, Portugal

**Keywords:** aquatic sensing platform, unmanned surface vehicles, multi-technology network systems, embedded systems in Internet of Things (IoT) platforms

## Abstract

The Internet of Things (IoT) is a rapidly evolving technology that is changing almost every business, and aquaculture is no exception. In this work we present an integrated IoT platform for the acquisition of environmental data and the monitoring of aquaculture environments, supported by a real-time communication and processing network. The complete monitoring platform consists of environmental sensors equipped in a swarm of mobile Unmanned Surface Vehicles (USVs) and Buoys, capable of collecting aquatic and outside information, and sending it to a central station where it will be stored and processed. The sensing platform, formed by the USVs and Buoys, are equipped with multi-communication technology: IEEE 802.11n (Wi-Fi) and Bluetooth for short range communication, for mission delegation and the transmission of data collection, and LoRa for periodic report. On the back-end side, supported by FIWARE technology, an interactive web-based platform can be used to define sensing missions and for data visualization. Results on the sensing platform lifetime, mission control and delay processing time are presented to assess the performance of the aquatic monitoring system.

## 1. Introduction

Aquaculture is one of the flourishing sectors in the food industry. Global fish production peaked at about 171 million tonnes in 2016, with aquaculture representing 47 percent of the total. With capture fishery production relatively static since the late 1980s, aquaculture has been responsible for the continuing impressive growth in the supply of fish for human consumption [1]. Continuous monitoring of the physical, chemical and biological parameters of aquaculture environments helps not only to predict and control the food production, but also to avoid environmental damage and the total collapse of the production process.

The monitoring of aquatic environments used to be highly inefficient, time-consuming and with a huge dependence on human-intensive field measurements for data collection. Without the proper use of technology, feeding activities, water analysis must be done in site with hand-held devices or collected for laboratories for further analysis [2]. With this process, the detection of any abnormal condition in the water quality usually happened too late, incurring in expensive costs and complex techniques to revert and stabilize the water quality [3,4].

However, with today’s advances in technology, it became possible to improve the monitoring process, addressing a larger number of problems related with accessibility, inadequate measurements or even temporal and spatial scales, increasing the productivity and minimizing the losses by constant monitoring of water quality parameters. Among the technologies that can leverage the monitoring process of aquaculture are the Unmanned Surface Vehicle (USVs) and the Wireless Sensors Networks (WSNs). The first one grants the possibility to collect aquatic samples without the need of human involvement on-site [5], while the second one establishes a communication infrastructure, critical for the transmission of information between the collecting units and the platform owners [6].

In this context this work proposes an end-to-end mobile solution for real-time monitoring of aquaculture environments. The complete solution includes the data collection equipment, formed by a fleet of USVs equipped with aquatic and outside environment sensors and several communication technologies and, on the back-end side, near shore gateways to receive, store and process the information transmitted by the USVs. Buoys with sensing and transmission capabilities are also included and integrated in the monitoring platform. A web-based platform is used to manage the USVs fleet and monitor the water quality. To alleviate the human-dependency on the data collection process, the aquaculture manager is able to set a list of environmental collection missions (which may include a list of locations and sensing parameters) that will be transmitted to the USVs fleet. With respect to the communication technologies, an Intelligent Network System has been developed, which includes IEEE 802.11n, usually denoted by Wi-Fi, Bluetooth and LoRa. The first ones are used in USV-USV, USV-Buoy and USV-Wi-Fi gateway communications for mission delegation, mission control and the transmission of data collection, while LoRa, due to its duty-cycle restriction, is making use of its long range capabilities for periodic report between the USVs and the LoRa gateway, for mission collection and other important requests.

The remainder of this paper is organized as follows. Section 2 discusses the related work. Section 3 describes the platform architecture, the functionalities and the Wireless Sensor Kit used. Section 4 presents the multi-communication technologies integrated in the network system, while Section 5 describes the missions’ assignment. Section 6 describes the methods implemented in order to acquire data in the platform. Finally, Section 7 presents a case study and discusses the performance results and Section 8 enumerates the conclusions and future work.

## 2. Related Work

Many monitoring platforms have been developed for aquatic environments. They can be classified as static or mobile, with single or multiple sensing units, or even with a single or multiple communication technologies. The authors of Reference [7] present a platform to measure pH and temperature levels in a fish farm, with a master slave architecture in a centralized approach. The sensing unit collects and transmits the information to a central unit through a wireless protocol. The central unit manages the network and stores all the received data. The wireless transmission follows the standard IEEE 802.15.4 and implements the routing protocol based on the ZigBee standard. The work in Reference [8] presents an aquatic drone equipped with a Raspberry Pi connected to an array of air and water quality sensors for quality monitoring. For air quality measurements, temperature, humidity and gas sensors were considered, while for water quality monitoring water, temperature and conductivity sensing channels were implemented. The collected data is stored on the drone’s computational platform and synchronized with a remote server database. In a similar way, the works in References [9,10] present the prototype and proof of concept of a distributed monitoring system for aquaculture. The solution is capable of monitoring the water quality based on static wireless sensor networks, without any type of movement, and therefore, mission control. All these works are characterized by the lack of movement with respect to the sensing devices and considering a single communication technology.

With respect to USV platforms for aquatic monitoring, several works have also been proposed. In Reference [11] the authors present a solution to monitor and quantify the pollution in the water produced by the feeding activities in marine fish farms. This platform uses an Underwater wireless system network in order to distribute the data to the nearest sync with a single communication technology. In order to improve the efficiency of the data sent, this platform can perform data fusion. In Reference [12], FLEXUS is presented, an affordable USVs’ swarm for water monitoring, imaging and air monitoring. This platform uses multiple Wi-Fi technologies (2.4 GHz and 5 GHz). Similarly, AquaBotix proposed the SwarmDiver [13] platform, where a team works as a single coordinated entity, while communicating with each other. However, it only includes pressure and temperature sensors and uses only one communication technology.

The work in Reference [14] proposes a heterogeneous Unmanned Aerial Vehicles multi-system, where a mechanism based on dividing the area between vehicles is implemented to increase the coverage area. By using a network and its protocols, the team can build up navigation information. In Reference [15] a communication-aided navigation mechanism platform is presented, where periodically, through a mobile *ad-hoc* network (MANET), all robots share their navigation tables. The work in Reference [16] proposes a heterogeneous cooperative platform composed of USVs and aerial drones to monitor the degradation indices and water parameters of a dam. The platform performs a mission with a pre-defined mission waypoint, delivering in real time the messages by radio in Line-of-Sight to the Command and Control Station or with the help of an aerial drone acting as a relay performing a mesh network. The platform cooperative builds a 3D map of the area. Again, these works only consider one technology and strict planned missions.

Regarding the motion planning algorithm used in USVs aquatic monitoring platforms, the work in Reference [17] proposes a motion planning algorithm in a fleet of commercialized USVs for aquatic monitoring which implements an offline mechanism to generate trajectories taking into account the connectivity to the ground station. In Reference [18] the authors present a decentralized platform of Autonomous Aquatic Crafts (AACs) for research on swarm algorithm, that comprises two networks, one internal and another external to monitoring the state of the AACs. Both networks are defined based on an *ad-hoc* wireless technology with Xbee. Several structures have been proposed to maintain the connectivity within a team. Approaches like leader-follower [19,20,21,22] and the virtual structure approach [23,24,25] are categorized as centralized control. Moreover, behavior-based structure approach [26,27] is categorized as decentralized control. Compared to these works, our approach is able to deliver an end-to-end solution, with a heterogeneous communication approach, motion planning based on the networks requirements in a swarm environment, in a self-organized way, coordinating both missions and data collection in a dynamic approach and providing a user-based web platform for visualization and coordination.

## 3. Aquatic Platform

The proposed platform, illustrated in Figure 1, is comprised of several teams of USVs and buoys that: (1) self-organize in the water through a mission and path planning approach; (2) communicate between themselves to control the mission and path and to never loose connection; (3) sense environment information both from the water and the air; (4) and communicate this information to a chosen Data Gateway Station (DTS) on land. These DTSs are connected to a Server in the cloud that stores the sensed information and the positioning of the USVs during the mission, which are then used by end-user applications such as a real time dashboard, alert platform and a mission control platform.

The system is quite versatile and can be used not only to aquaculture applications but also in environmental monitoring in coastal, estuaries and lagoons environments. The system was designed to retrieve the frequently measured parameters that aquaculture operators collect with manual probes or in fixed points and it is highly suitable for aquaculture applications, especially in earth ponds of large dimensions.

**Buoys** are stationary stations, usually without wired connectivity, composed of a set of environmental sensors, with the purpose of collecting environmental information that can be used to control scenarios such as aquaculture.

The Mission Controller (MC) is a fixed network element on land that gives the orders to each Unmanned Surface Vehicle (USV) to start a mission/task. It receives these orders through a user platform and sends them in the right format to the USV. To be able to access all USVs in the water, through a larger connectivity range, the MC uses the LoRa technology (https://lora-alliance.org) to communicate with the USVs.

The **DTSs** are also fixed network elements. These nodes act as endpoints connected to servers in the cloud, since they are the final elements of the data dissemination plan. Due to the support of multiple communication technologies in the water between the USVs and between the USVs and Buoys with the DTSs, the gateways will be endpoints for both Wi-Fi and LoRa communications.

**USVs** are mobile entities that work within a team. These nodes have both Wi-Fi and LoRa communication capabilities. With the Wi-Fi technology, the communications between elements are assured by a Delay Tolerant Network (DTN), where these elements act as relays in a store-carry and forward fashion. These mobile nodes are responsible for forwarding and collecting the sensing information until it reaches the DTSs. With the LoRa technology, since it is able to reach 2–6 Kms, the communication to the DTSs is direct between the respective USVs and the DTSs.

The **Server**, has the responsibility of receiving all the data from the clients and then storing it into a corresponding database for both real-time and posterior use in several applications, both mobile and web. The platform implements a client-server architecture, using a FIWARE (https://www.fiware.org/) broker for interoperability between different systems with a fast performance. A client can submit information using REST requests which is then automatically stored in a MongoDB database (https://www.mongodb.com/).

Another FIWARE module is used to store all the data that comes through the FIWARE broker called Cygnus (https://fiware-cygnus.readthedocs.io/en/latest/). This module automatically persists all the data in a PostgresSQL database (https://www.postgresql.org/), creating a new table if a new model appears, with the same table structure for all models. This database exposes an endpoint that can be used to watch and retrieve the sensor data.

The aquatic platform uses several communication technologies allowing aquatic elements to maintain communications with at least one DTS with short or long range communications and thus providing the connection to the server for a longer time. LoRa plays an important role in the platform because, when any USV loses connection to the gateway over Wi-Fi, this is the only communication interface that can reach the land while maintaining the monitoring task.

### 3.1. Functionalities

The main purpose of this platform is to sense and process data from the sensors in the USVs and Buoys and send it through the DTS to the server, which is accessible to aquaculture workers and useful for researchers to improve or maintain the network and aquaculture performance. With the data collected, this platform is used for data analysis, and in an enterprise context, it can be used to prevent sub-optimal welfare and mortality or unwanted events, or even to control the automatic feeding portions of each tank.

In terms of navigation, the platform is capable of completing a mission or task as a team while maintaining connectivity between each USV. It allows the implementation of machine learning algorithms to improve the team’s organization. Due to the connectivity to a DTS, this platform can assign new missions at any time and from everywhere to each USV via the MC.

Moreover, the platform is connected to a mobile App, that can be used to collect data from the device and alerts the user for a specific trigger inserted. The mobile app also allows for the visualization of the data collected.

The Web dashboard allows for a map visualization of the data, showing every location where data was collected and allowing the user to watch the most recent values. It also contains graphs that show the evolution over time of any particular sensor. Using the dashboard, it is possible to define threshold values that, when toppled, a notification is sent to the dashboard. Mission routes are defined also using the dashboard, with the locations defined using a map. After being created, they can be started and also stopped, if they are still in queue.

This platform is also scalable because it can admit a change in the number of devices without changing the code or configuration files and it is robust as it can recover and still complete the mission when a vehicle malfunctions or shuts down.

### 3.2. IoT Wireless Sensor Kit

#### 3.2.1. Prototype Setup

The design of the system USV is shown in Figure 2. The designed model is divided into four main parts, as described in Figure 3, which are the data acquisition, the central processing unit, the communication module and the power unit. The hull of the USV is built using ABS and the base of the buoy is made of styrofoam and plastic. The electronic devices for aquatic monitoring are installed inside the hull or in a water-proof box in the buoy.

#### 3.2.2. Electronic Setup

Due to the fact that there is no commercial sensor kit suited to the system’s needs, the entire sensor kit was designed from scratch, implementing hardware-appropriate firmware. Each USV, DTS and MC is composed by a Raspberry Pi model 3b+, as illustrated in Figure 4a, with *buster* Raspbian OS (https://www.raspberrypi.org/blog/buster-the-new-version-of-raspbian/) and ROS Melodic (https://www.ros.org/). Moreover, the Buoy (illustrated in Figure 4b) is composed by a Compute Module Raspberry Pi version 1, to be able to accommodate more sensors, due to the expanded IO. The USV’s also have space to add more sensors, but the bottleneck is the central processing unit capabilities (memory and power). All devices equipped with LoRa communication also includes a SX12772 LoRa module and Multiprotocol Radio Shield manufactured by Libelium [28].

In order to lower the cost of each device, each USV has different types of sensors, instead of each one having all sensors integrated. A printed circuit board (PCB) was designed, as illustrated in Figure 5, to minimize mechanical requirements and facilitate the assembly process. However, this solution makes the platform vulnerable, due to the unprotected connections. Figure 5 represents the different devices that can be assembled using the same PCB, easing the process of changing to different sets of sensors, not changing any connections or code. Moreover, the Buoy also integrates another PCB to connect from the compute Module IO to the Raspberry Pi 3 IO.

Each PCB can integrate the sensors presented and described in Table 1. The Liquid Level sensor and the Dissolved Oxygen are only used in Buoys and in these devices we do not use the Ultrasonic sensor to measure obstacles. Each device has one power bank to feed the Central Processing Unit and the sensors. However, the USVs have another battery directly connected to the motors. Figure 2a shows that, close to the motors there is a LED light and a switch on opposite ends. The LED will turn on when the battery reaches critical levels and the switch acts as an emergency button to the motors. When flipped to the OFF position, the motors stop. The chosen hardware provides a low-cost device that has a modular structure, making it easy to repair.

### 3.3. Software Architecture

Figure 6 illustrates the modules of each device in the platform. The *Sensor Controller* module has two routines, one for reading sensors when requested from an external source or module and a routine where sensors are read periodically. Both routines can be configured by a configuration file. In an USV, there is a periodic routine for the navigation mechanism with a period of one second (minimum update rate of GPS). When the vehicle reaches the target, it makes five consecutive measurements and creates a data message of the average of each sensor. In an Buoy, only the periodic routine is enabled. Then, the data message is sent through a ROS topic to the *Path Planner* module, to update the state machine that controls the device and then it is sent to the *mobile Opportunistic VEhicular (mOVE)* module to be delivered to any DTS.

The *mOVE* module is a Delay Tolerant Network (DTN)-based architecture developed in our group [29]. It supports several technologies such as IEEE 802.11n (Wi-Fi technology) and IEEE 802.11p (Wireless Access in Vehicular Environments, which is not used in this platform). This module is responsible for delivering data messages to the server via Wi-Fi. It has a Store-Carry-and-Forward Mechanism to cope with the connection disruption that can occur. It integrates routing mechanisms, from the simplest ones, such as Epidemic, to the more complex ones based on connection link quality, social heuristics or others. It also integrates a neighbouring table, so each device has the entire network information spread among devices by advertisement messages. When there is no connection to any Wi-Fi DTS, then this module forwards the *Data messages* to the *LoRa* module.

The *LoRa* module is responsible for handling the delivery and reception of LoRa packets. Once a *Start Mission message* is received, the same will be forwarded to the *Task Provider* module to process and deploy the mission. Additionally, this module listens the *mOVE* module for Sensor Data payloads that needs to be structured with the LoRa header and dispatched to the LoRa DTS. When the *LoRa* module receives a *Start Mission message*, it forwards it to the *Task Provider* module to store the information until it is time to start the mission. When the mission has to start, the *Task Provider* sends the navigation information to the *Path Planner* module. Each time this module receives a new task, it stops the navigation mechanism of any old task still in motion (not deleting data packets) and starts the new process, with a new grid map, targets and nodes.

The *Path Planner* module implements single and team navigation mechanisms. The single navigation mechanism is responsible to allocate targets regarding the type of sensors each vehicle has and the requirements of the mission and to find a path to that specific target avoiding static obstacles. The team navigation mechanism is responsible for maintaining connectivity between the team vehicles and is responsible to avoid collisions between them. These mechanisms are described in more detail in Reference [30].

The *Bluetooth* module is responsible for saving the last gathered values in a database. When a Bluetooth device with our App is near and chooses the USV, the USV starts to directly deliver all the database information to that device by Bluetooth technology.

## 4. Network Communication System

The proposed platform has the capability to provide communication between devices by using simultaneously multi-communication technologies in a synchronized approach. The communication technologies in place are: LoRa, IEEE 802.11n (Wi-Fi) and Bluetooth, as illustrated in Figure 7.

**LoRa:** this is a long-range wireless communication system aimed to be used in devices where energy consumption is of utmost importance. This technology grants high communication ranges up to 6 Kms in our previous tests [31]. However, it provides low data-rates, complying with a very strict duty-cycle regulatory restriction. The regulations in the ETSI EN300-220-1 document establish diverse requirements for short range devices, mostly on radio activity, in which radio transmitters must adopt duty cycled restrictions. Generally, the transmitters are constrained to 1% duty-cycle, which means 36 s of use per hour, which is sufficient to meet the communication needs of the system.

The platform uses small time windows, commonly five minute windows. This time window has been selected to ensure that, depending on the Time on Air (ToA) of data packets (which depends on the packet size and LoRa communication mode), each USV, Buoy and Mission Controller can send more than one data packet per window. In this way, it is possible to transmit multiple data packets in the same transmission window granted to a sending node, thus reducing the number of multimedia accesses per node. The LoRa communication mode for the devices is configured with the following properties: BW = 500 Hz, CR = 4/5 and SF = 12. Moreover, three seconds of the available window duty cycle allows a node to transmit more than 300 bytes. Therefore, as mentioned above, it is possible to transmit multiple data packets along with the required control packets.

Each device capable of communicating through LoRa has a module responsible for handling the action of receiving and sending LoRa packets in a synchronized manner. This module takes into consideration the LoRa duty-cycle already mentioned above and it is set up in a way that it can sink a number of packets from the sending queue where the total sum of their length does not surpass a specified length, usually between 300 bytes and 400 bytes. Consequently, the LoRa communication will not be able to transmit packets through LoRa medium for at least 5 min approximately.

The LoRa Libelium SX1272 module has a support library that manages the SX1272 LoRa module. The maximum length of a LoRa packet is 255 bytes, 5 of them are used by the fields described above. The LoRa packet payload includes a header of 8 bytes and the remaining 242 bytes are used to carry data payload. The header consists of the destination node address, source node address and packet type, as specified in Figure 8.

The MC sends a client request aimed to schedule USV missions, as well as to notify USVs to update their missions and furthermore update their sets of geographical points. Each request is originated by a client, is sent through the server and reaches the MC in a JSON Format, where it is mapped the following information: the type of the request, the numerical identification of the destined USV and a specific byte stream payload corresponding one of the illustrated in Figure 8. Having this data, the MC is able to parse the request and build a packet with a specific format, such as the shown in Figure 8, aimed to be used by packets that are sent through LoRa.

Each USV’s LoRa module works according to a MAC protocol for Multi-Gateway in LoRa based on Ready to Send(RTS), Clear To Send (CTS) and Control Clear to Send (CCTS) messages in order to deal with a multi-gateway environment, to ensure that only one of the DTS is ready to start receiving data packets originated by one of the USV [29]. For a node to be able to send data packets through LoRa, it must wait for a DTS to initiate a session, according to the mentioned protocol, so it can receive the LoRa packets. An USV or Buoy listens for the two types of packets mentioned, sent through LoRa to either start a mission (with Start Mission payload) or to one of the nodes reaching the DTS and update the missions, which corresponds to the packet with the Request Update Mission Payload.

Sensory data payload is structured in a sequential way, where it is specified the ID of the sensor following the values sampled by the respective sensor (Figure 9). Some of the sensor devices used by a node acquire more than one type of value (for example, the GPS, where latitude, longitude and speed are measured), thus having more than one sensor value field on the structure of the packet’s payload. This packet structure is variable according to the available sensors composed by the USVs.

USVs will transmit their sensory data packets through LoRa in case that the same packet cannot reach its endpoint through *mOVE*, i.e using Wi-Fi and the DTN instead of LoRa. When this happens, the LoRa software module is notified, a LoRa packet is created and sent to the DTS once a session is established.

The DTS also works according to the MAC protocol for Media Access Control for Multi-Gateway in LoRa. This means that it establishes a LoRa session with a LoRa node when it is requested, as long as the DTS is free to do so. The received LoRa sensory data packets are parsed and sent to the server.

**Wi-Fi:** All nodes in the platform comprising Wi-Fi modules, meaning all USVs and Wi-Fi DTSs, have a DTN implemented. DTNs were designed for heterogeneous networks that may have connection disruption. This platform implements the *mOVE* architecture supporting IEEE 802.11a/b/g (Wi-Fi technology) and others, with a link quality-based routing strategy proposed in Reference [32]. This routing strategy has a replication avoidance mechanism and it has been proven to be better among other routing strategies for these types of scenarios.

**Bluetooth:** Each USV or Buoy is able to communicate through Bluetooth Low Energy (BLE) to a mobile phone with a mobile application, if this mobile phone is in reach of the USV and Buoy. For this purpose, each USV or Buoy has a dedicated module to manage a Bluetooth session. Once the session is established with an endpoint device (the mobile app in the mobile phone), the module will gather the last sampled sensory values stored at a database and will share it with the client. If the session is still established, i.e the client is still in reach through Bluetooth and new sensory values are sampled, the USV or Buoy transmits this information to the client and stores it at the database in the case a new client comes up and starts a new session. Each packet is built in a JSON format, where sensory values are mapped with the correspondent sensor identification. This application is an android mobile app dashboard dedicated to establish a session, such as the mentioned above with an USV or Buoy, capable of parsing and processing the received values, displaying them through graphs.

## 5. USVs’ Missions

A mission represents a set of geographical points, tuples of values representing the latitude and longitude. One of them is explicitly the starting point of a mission. Each mission has a specific identification in order to indicate which of the missions are requested to deploy. The group identification of the sensors to be used in the mission is also specified. Finally, a mission mentions which nodes are allocated to that specific mission.

In order for a node to initiate a mission, first it needs to have the information of the assignment. The USV stores all relevant mission data in order to correctly deploy the assignment when required and allowed it to do so. When a LoRa start mission packet reaches a USV, the USV selects the records of the specific mission from the database, checks if it is assigned to it and starts the mission at the specified time, which is included on the start mission packet payload.

Because the mission’s data is usually extensive, it is not appropriate to send it through LoRa, where the maximum length of a LoRa packet is 255 bytes, as mentioned. Instead of using LoRa technology to inform the USV about the available missions, so it can populate its database with data records of the missions of interest, the MC sends a *Request Update Mission* Message through LoRa, to notify the USVs that they need to update their missions’ data. This request contains also the location of the DTS.

Once a USV reaches the area where the DTS is placed, it tries to communicate with the DTS through Wi-Fi using *mOVE*. The USV gathers the data from the DTS and the same shares the information with the other USVs in reach. This is done in order to avoid all USVs to proceed to the DTS geographical area. By the time the mission information reaches the USVs, it is stored in their databases. The DTS retrieves the missions’ data from the server so that it can send it to the USVs. When a DTS discovers a new neighbor node through the *mOVE* module, if it does not contain the mission, the missions’ data (the *Start Mission* Message) is fetched through a HTTP GET Request to the server, which comes in a JSON format. Then, this information is sent to the USV through Wi-Fi. This process is illustrated in Figure 10.

With the mission information received, the USV will start the mission at the time specified in the payload of the mission. Once the mission starts, the USVs will start the correspondent mission at the specified start position. From there, they will travel through a set of points defined by the mission, collecting environmental and aquatic data just after it reaches a point of interest and send it to the DTS using LoRa or Wi-Fi (through *mOVE*) technology. This process is illustrated in Figure 11.

## 6. Data Acquisition and Analysis

To manage the data in the server, FIWARE is used. FIWARE is an universal set of standards for context data management (www.fiware.org/developers/). It is used together with its several allied components, like the FIWARE Context Broker, in order to manage data that is dynamic and always changing. If there is a new sensor to be added, FIWARE will accept it without any modifications or configurations needed. To use the aquatic and environment data with FIWARE, a data model adhering to the FIWARE rules had to be created. We created our own model, called “Aquatic model”, by borrowing attributes from FIWARE own’s Weather Model and Water Quality model where possible and creating other properties where needed. The aquatic model is described as follows:**id:** Unique identifier**type:** Entity type. It must be equal to Aquatic.**dateObserved:** The date and time of this observation in ISO8601 UTC format. It can be represented by a specific time instant or by an ISO8601 interval.ted by a GeoJSON geometry.**refDevice:** A reference to the device(s) which captured this observation.**temperature:** Air temperature.**humidity:** Air’s relative humidity observed (percentage).**atmosphericPressure:** The atmospheric pressure observed measured in Bars.**pH:** Acidity or basicity of an aqueous solution.**uvIndex:** Incidence of UV light.**ultraSonic:** Underwater distance to objects.**underwaterPressure:** Underwater pressure observed measured in Bars.**waterTemperature:** Water temperature.**conductance:** Conductance of the water.**turbidity:** Water’s observed turbidity percentage.**depth:** Observed depth at the sensor unit location.**voltage:** Percentage of battery left.**O${}_{2}$:** Ratio of O${}_{2}$ in water.**heading:** Direction of the USV.

Before the data is ready to be sent to the server, it needs to be processed and turned into a JSON message that is compliant with the Aquatic model. The received message is unpacked and a dictionary is created with each entry being a different sensor reading. Then, a JSON message will be created with the available sensors in the format of the Aquatic Model.

This message is then sent into a Mosquitto broker (https://mosquitto.org) that is running locally in the DTSs. The Mosquitto broker then has a bridge connection to the FIWARE broker in the server, so that the messages in the Mosquitto broker are automatically redirected to the FIWARE broker when there is an available connection.

The FIWARE broker automatically stores the information of an entity (entity is an instance of a data model, in our case the Aquatic model) in a MongoDB database (https://www.mongodb.com). The downside of this database is that, when new information arrives for an entity, it overwrites the previous information, leaving no historic of data. Since we want to be able to access all data gathered by our USV, not just the latest data, we use another FIWARE component called Cygnus. This component will be responsible for persisting all the data received in a PostgresSQl database. It creates a new table for each Data Model, where each line is an attribute of the data model, its value and the associated timestamp. It is from this database that we retrieve the data later for display in the dashboard and for analyzing in the alarm system.

The alarm system consists of a component that is continuously analysing the data and checking if any value is abnormal, with abnormal being “too high” or “too low”. The values for “too high” and “too low” are defined by the user. Using the dashboard, the user will be able to define threshold values for all sensor types which will then be used by the alarm system. When an abnormal value is detected, a notification is generated in the dashboard to warn the user. Optionally, an email can also be sent.

### 6.1. Automatic Indirect Data Acquisition

To visualize the data, the user can use a dashboard where a map will be presented with various markers on it (Figure 12). Each marker will represent a location where the USVs have gathered data. By clicking on each USV, the most recent values of each sensor reading will appear below the map. Alternatively, it also shows graphs depicting the change of a particular sensor over time.

The user will be able to define new missions for the USVs using the dashboard. In the route definition page (Figure 13a), the user is presented with a map where he can click to define a way-point where the team should make readings. In this menu, the user also has to define a number and a name for the mission, which will later be used when he wants to start or stop it. To start a mission, the user should head to the *mission select* page (Figure 13b), where he can select it from a list of all routes available and specify a time and date for the mission to occur.To cancel the mission, the user should head to the *mission information* page (Figure 13c) and eliminate the mission to cancel.

### 6.2. Direct Data Acquisition

The option to gather data directly from each USV is also available, using a mobile app. The user may go directly to the USV and use the mobile app to transfer the data from the USV to the mobile device, through Bluetooth. All data can then be watched on the mobile app itself, as can be observed in Figure 14.

The mobile app is created with the purpose of collecting the values in real-time obtained by node’s sensors, once a Bluetooth session is established. The goal is to exhibit the values obtained by each node’s sensor relatively to the acquired time. This is done through iterative graphs per node’s sensor. The dashboard is developed in order to display the node’s sensory data sent through LoRa.

Both modules suffered modifications throughout this study in order to receive and interpret correctly the system’s message formats sent by the USV and buoys of the system in study.

## 7. Case Study

This section describes the deployment of the platform in the aquaculture environment and the obtained results from the running use case.

### 7.1. Full Deployment

This platform can be used in several aquatic environments. Table 2, maps the sensors to the use in these several environments. The system was designed to retrieve the frequently measured parameters, described in Table 2, that aquaculture operators collect with manual probes or in fixed points. Other sensors could be used and integrated in this platform. However, this solution was developed with earth ponds or earth tanks in mind, where water circulation is forced (one in and one out), and therefore, does not require a water flow sensor. Moreover, there is no system, that continuously measure toxins and pollutants, available on the market. The normal process to measure those parameters includes the collection of a water sample, followed by the analysis in a lab to screen for toxins and or pollutants.

The biggest advantage of this system is the ability to dynamically map the area instead of depending on fixed measuring points that can mislead the technicians in charge and the alarm systems. However, this solution has limitations in indoor environments: as a result of the use of a GPS sensor in the navigation system, the mobile nodes can only be used in outdoor environments. Moreover, due to the size of the USVs, this platform is not suitable for open water problems.

There are three types of platform’s setups:Static deployment with a Buoy;Team deployment **without** missions’ data update;Team deployment **with** missions’ data update.

For the setup number one, we need to deploy a LoRa DTS connected to the server, along with the Buoy. For the other setups, the platform has to have a LoRa DTS and a MC and can have a Wi-Fi DTS. If we only have a LoRa DTS, then the missions’ data cannot be updated, entering the setup number two. If we have at least one Wi-Fi DTS reachable to the team, then the platform is in setup number three and can receive new missions’ descriptions.

In scenarios where the missions’ descriptions do not change often and the data being measured has a low periodicity, the inclusion of a Wi-Fi DTS may not be the most efficient compared to only having a LoRa DTS. For example, in AlgaPlus (https://www.algaplus.pt/)’ earth ponds (with large dimensions), the most suitable platform setup is a team of USVs with a LoRa DTS connected to the server and a MC to initiate the mission on a specific time. Moreover, the Buoy, when connected to the battery, it immediately starts to measure the variables periodically. However, the USVs only start to move and measure after receiving a mission, through a request to update missions’ data, or another mission.

### 7.2. Experiment

This section presents a case study used to assess the performance of the platform with respect to the energy autonomy, connectivity, processing delay and mission delegation. The routing protocol and mechanisms required to assure the USV-USV communication by the means of a Delay Tolerant Network were already evaluated in Reeference [32] and they are out of scope of this work. Moreover, the navigation algorithm was evaluated in Reference [30].

The goal of the evaluation in this paper is to analyze the battery capacity, the connectivity between the gateway and the USVs, and among USVs, as well as the time spent in the information processing task by the USV. For that, the scenario illustrated in Figure 15 was considered, comprising two USVs, labeled as 51 and 52, both equipped with the same sensors, three targets, one DTS with Wi-Fi and LoRa, labeled as 31 and one MC.

As stated before, each USV has two batteries, one for the Raspberry Pi and another for the motors. The life of the motor’s battery was found to last for 75 min, when the motors were running continuously at maximum speed, being this the worst case scenario.

In this experiment, every USV starts with no information about any mission. Due to the location of the USVs, the team is in contact with a DTS in the beginning of the experiment. Automatically, the DTS starts sending a missions’ data message to each USV directly connected. When an USV receives this packet, the database is populated with the newest information of the existing missions. All results comprise the mean and 95% confidence interval of 5 experiments in the same conditions.

Figure 16 presents the delay and processing time in each USV of a *Collect missions’ data* Message and a *Start Mission* Message. These messages take between 1 and 2 s to arrive in the USV and be processed. This shows that, for this scenario, a Start Mission message shall be sent with a starting time higher than the message timestamp plus 3 s (the delay and processing time of a *Collect missions’ data* Message plus the time of a *Start Mission* Message).

The task, in this experiment, consists in visiting three targets, taking the most advantage use of resources. When the mission starts, the team’s navigation mechanism assigns target A to USV 51 and target B to USV 52. When USV 51 reaches target A, it sends a message containing the values measured in A through Wi-Fi. Then, target C is assigned to USV 51. When USV 52 reaches target B, it also sends a message to the DTS and because there are no more targets, this USV waits for the mission to end, while maintaining connectivity to USV 51.

This mission takes on average 564 s to be completed (with a 95% confidence interval of 180.211 s), since the start of the mission advertised in the *Start Mission message*.

In this experiment, we wait until receiving all Sensor Data Packets, so as a result, we see a 100% delivery ratio. However, in *mOVE*, if the Acknowledgement message is not received within a specified limit, that packet is re-sent. In this experiment we have observed 5 re-sends in total out of 25 packets. On the target C, LoRa packets of 104 bytes were sent to the DTS. The total time spent sending LoRa packets in a LoRa session from a monitoring node to a DTS was approximately 4 s, pausing for 7 min before sending another packet. As a result, it is expected an USV to be able to send approximately 24 LoRa packets per hour with the mentioned sensor data payload.

To study the efficiency of the server when receiving data from several USVs, we flooded the gateway with packets to be transmitted to the server. Due to the DTS characteristics, that only sends a packet to the server every 100 milliseconds, we concluded that the server has 100% ratio of packets processed by packets received over time, being able to handle and process each packet under 100 milliseconds, having no packets left in the queue waiting at any time. On average, it takes the server 0.19151 s (with 95% confidence interval of 0.0302 s) to process and save the packet.

## 8. Conclusions

This paper proposed an aquatic monitoring platform capable of collecting environmental and aquatic data from specific geographical locations. This is performed by using a swarm of USVs and Buoys, combined with a set of sensors. The combination of the hardware and software modules of this platform is able to provide the following tasks: collect the values from sensors, execute USV tasks in a swarm, deliver the acquired environmental data through multiple communication technologies, and even be capable to request the information directly through a mobile app. The communication technologies comprise IEEE 802.11n (Wi-Fi) and Bluetooth for short-range communications, mission delegation and data collection transmission, and LoRa for periodic reporting.

With the support of FIWARE technology, an interactive Web-based dashboard has been produced to process and display the obtained environmental and aquatic data measured by the monitoring nodes. Through the same web application, an end-user is capable of creating missions, request the monitoring platforms to deploy specific missions and update their tasks.

The experimental results have shown that the platform achieves a robust dispatch of sensory data to the central station, either by sending it between the monitoring nodes until it reaches the DTS, with the support of the *mOVE* module implemented in each USV and Wi-Fi DTS, or by delivering the sensor data packets through LoRa. Moreover, the results have shown that it is reliable for the USV to deliver the collected data through the *mOVE* DTN, as long as the Wi-Fi DTS is reachable by the use of this approach. In the case of a wider distance between the USV and the Wi-Fi DTS, LoRa comes up as good backup to deliver the acquired sensory values to a LoRa DTS.

To overcome the huge dependence on human-intensive field measurements for data collection in aquatic environments, specifically in aquaculture environments, and the consequence degradation of the water quality responsible for the damage on the proper functioning of aquaculture environments, a system such as the one presented in this work might be very useful as a support platform for aquaculture farmers and workers. This system is able to acquire recent data of the water properties of aquacultures, that may be crucial to take appropriate intervention, allowing the workers to anticipate and mitigate the adverse events that may influence the quality of the water, thus avoiding bad consequences on the fish health.

As future work, the platform will be deployed in real aquaculture environments. Moreover, the support of multiple swarms will also be researched.

## Figures and Tables

**Figure 1 sensors-19-04695-f001:**
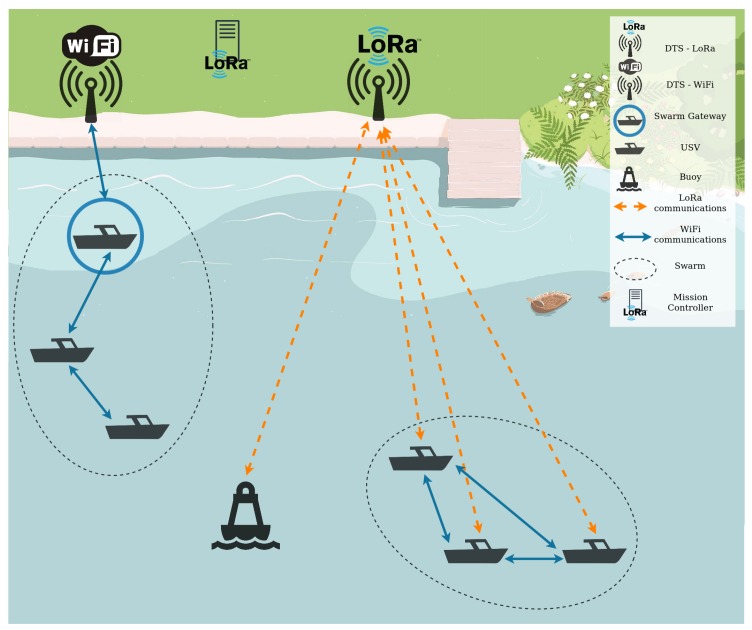
Platform overview.

**Figure 2 sensors-19-04695-f002:**
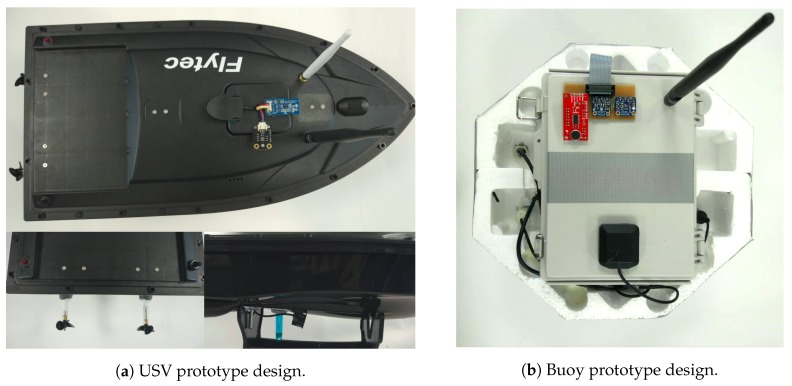
Overall prototype design.

**Figure 3 sensors-19-04695-f003:**
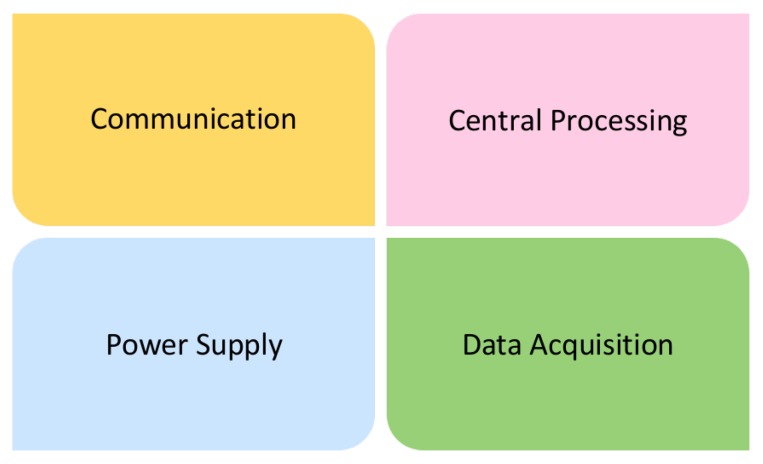
Architecture overview.

**Figure 4 sensors-19-04695-f004:**
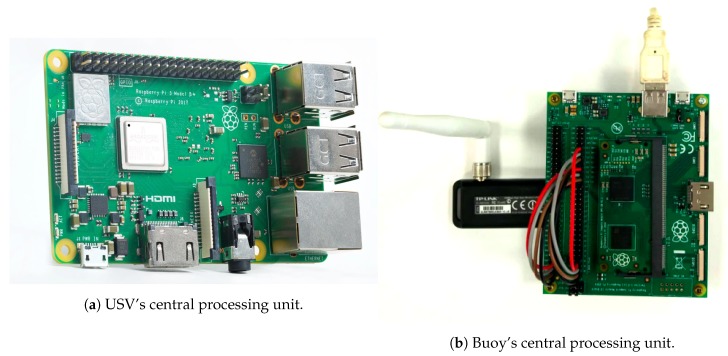
Overall prototype design.

**Figure 5 sensors-19-04695-f005:**
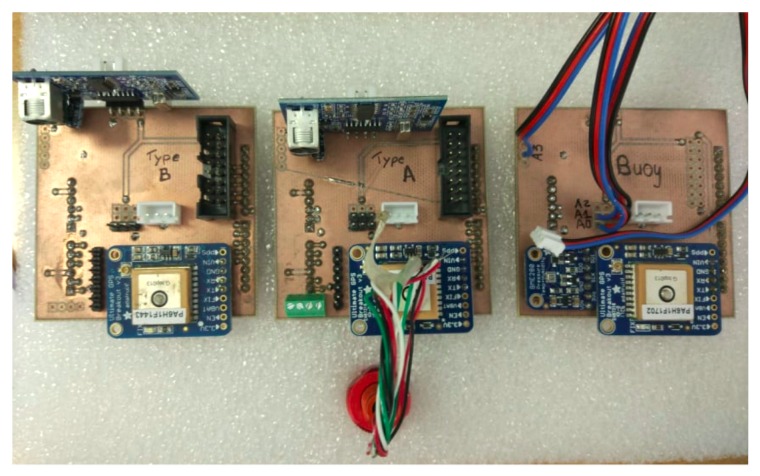
PCBs used in USVs and Buoys.

**Figure 6 sensors-19-04695-f006:**
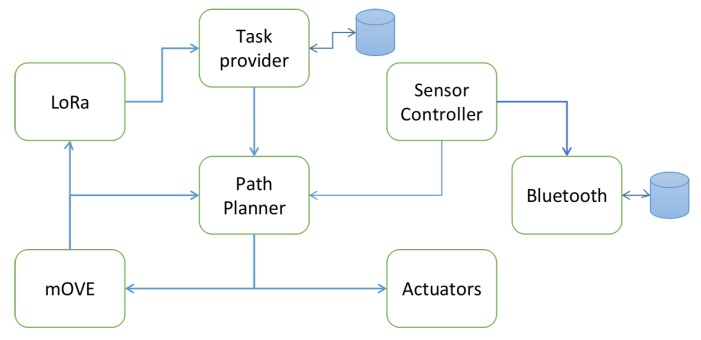
USV’s software architecture.

**Figure 7 sensors-19-04695-f007:**
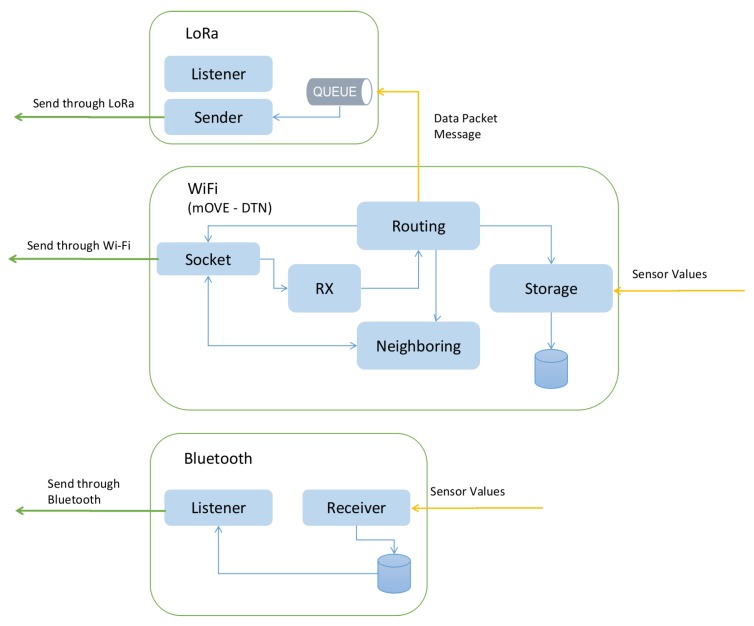
Multi-technology networking overview.

**Figure 8 sensors-19-04695-f008:**
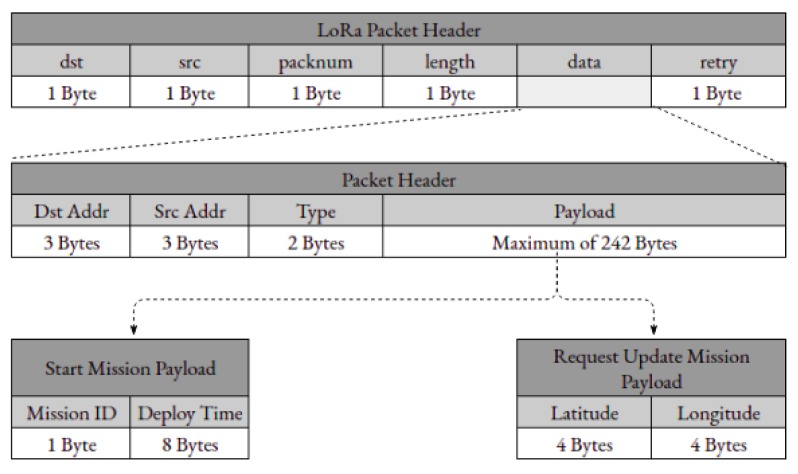
LoRa packet structure and two types of payload: Start mission payload and request update mission payload.

**Figure 9 sensors-19-04695-f009:**
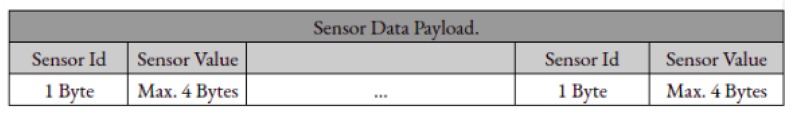
LoRa sensory data payload structure.

**Figure 10 sensors-19-04695-f010:**
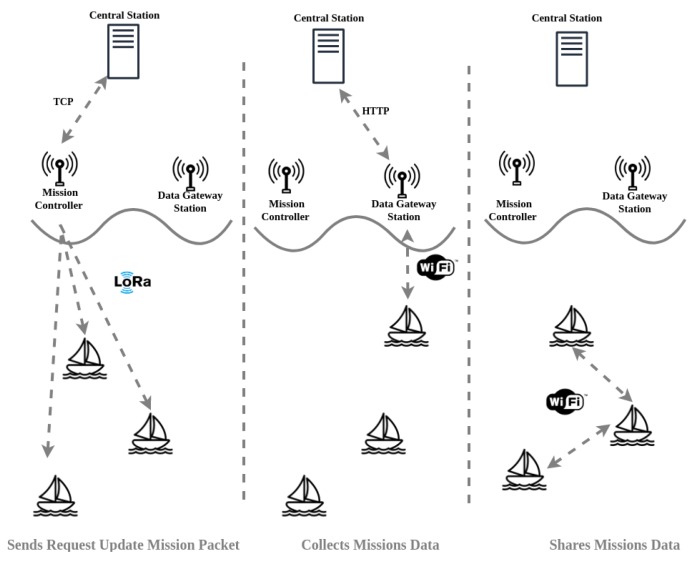
Mission update operation.

**Figure 11 sensors-19-04695-f011:**
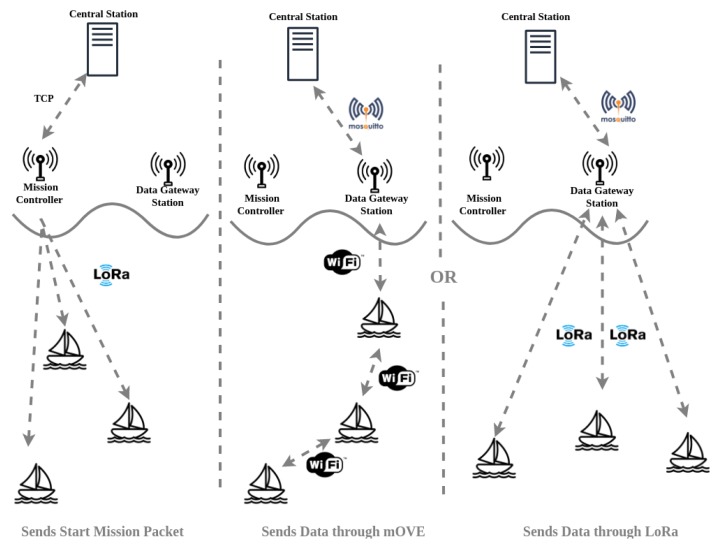
Mission start operation.

**Figure 12 sensors-19-04695-f012:**
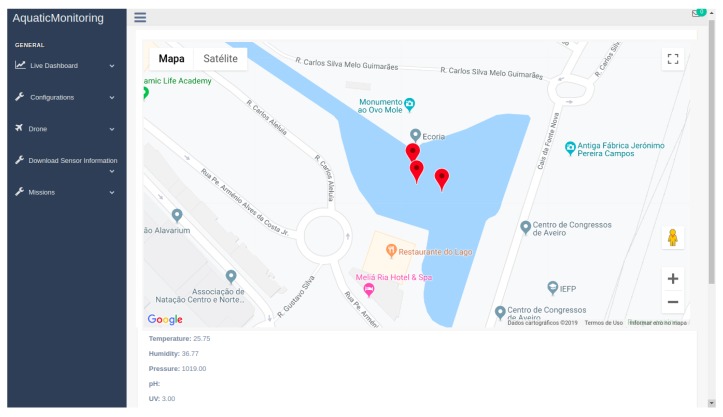
Dashboard presenting the information.

**Figure 13 sensors-19-04695-f013:**
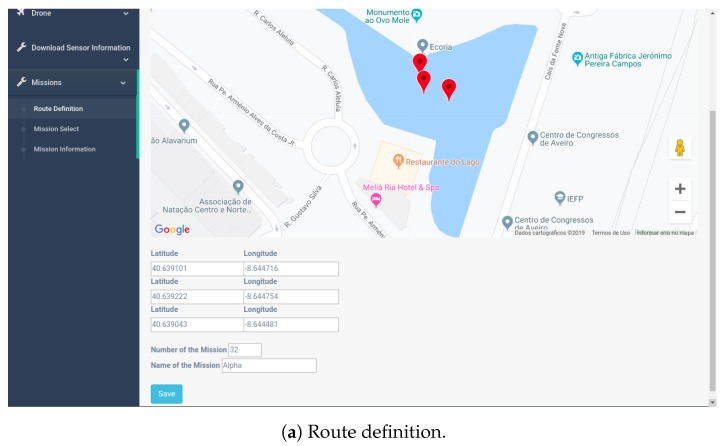
Mission-related information on the dashboard.

**Figure 14 sensors-19-04695-f014:**
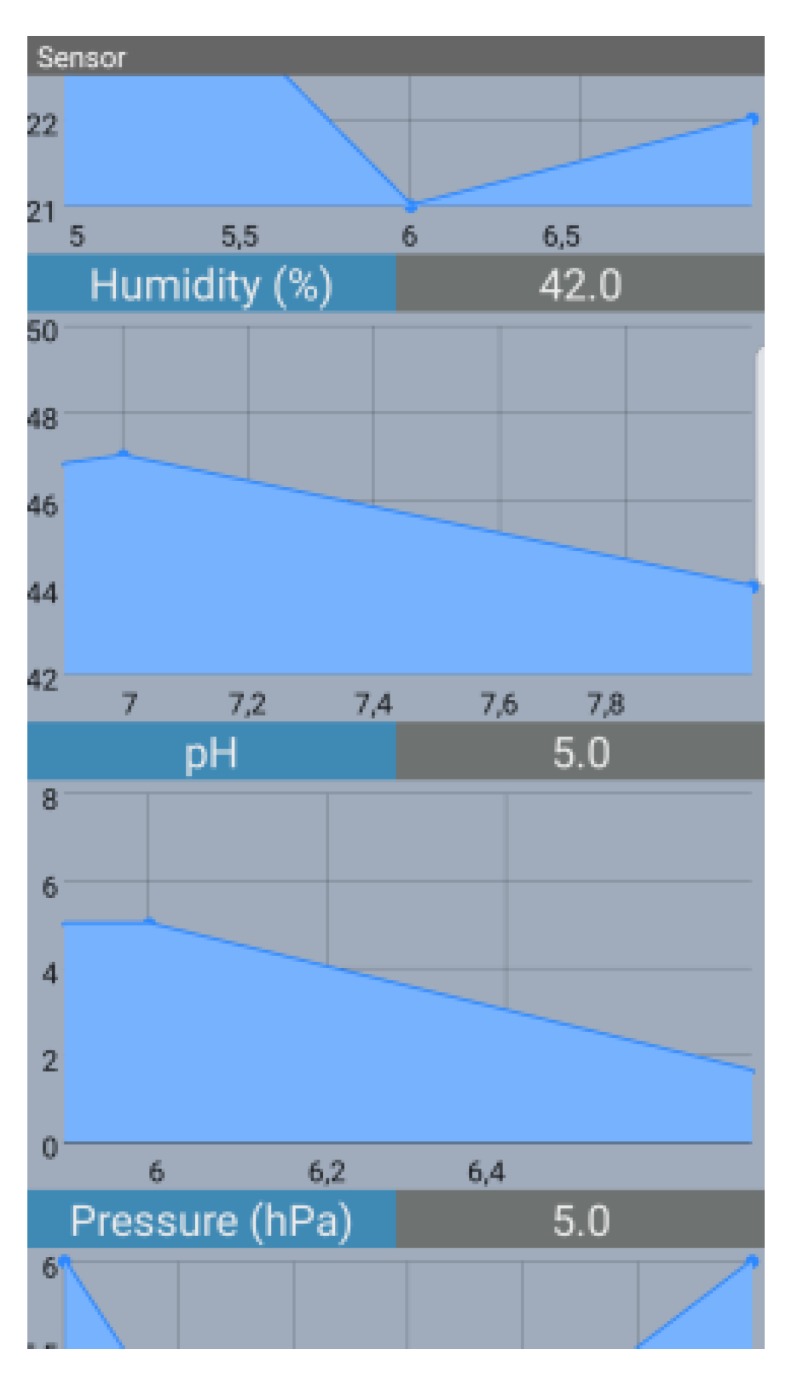
Mobile app.

**Figure 15 sensors-19-04695-f015:**
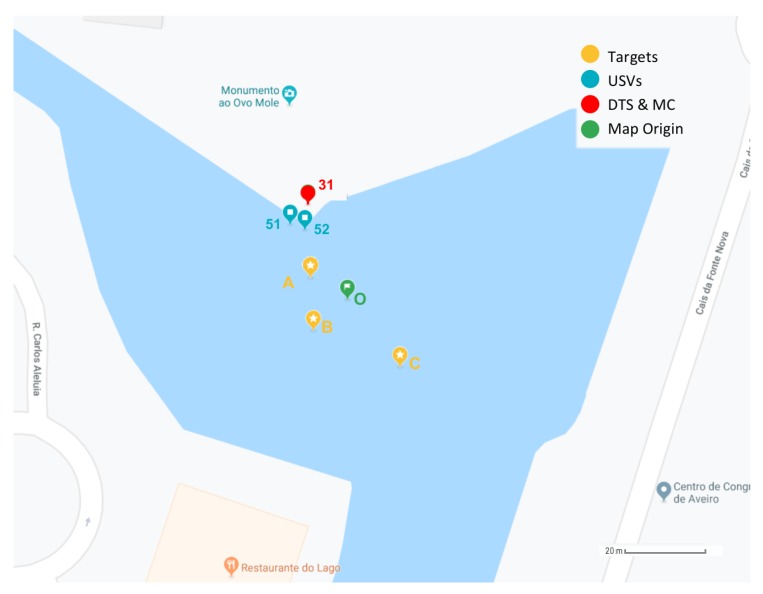
Case study scenario.

**Figure 16 sensors-19-04695-f016:**
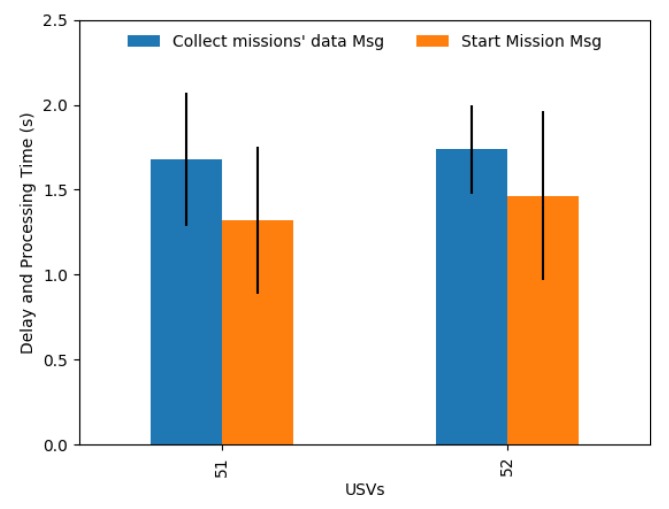
Delay and processing time of the *Collect missions’ data* message and the *Start Mission* message.

**Table 1 sensors-19-04695-t001:** Sensors description.

Name	Measured Parameters
Temperature-DS18B20 (https://www.adafruit.com/product/381)	Water temperature
Depth/Pressure-BAR30 (https://docs.bluerobotics.com/bar30/)	Depth and pressure
pH-SEN0161 (https://www.dfrobot.com/wiki/index.php/PH_meter(SKU:_SEN0161))	pH level
Conductivity Kit K1.0 (https://www.atlas-scientific.com/_files/_datasheets/_circuit/EC_EZO_Datasheet.pdf)	Electrical Conductivity
Turbidity-SEN0189 (https://www.dfrobot.com/wiki/index.php/Turbidity_sensor_SKU:_SEN0189)	Levels of turbidity
Ultrasonic-SEN0208 (https://www.dfrobot.com/wiki/index.php/Weather_-_proof_Ultrasonic_Sensor_with_Separate_Probe_SKU_:_SEN0208)	Distance (both depth and obstacles)
IMU SEN0140 (https://www.dfrobot.com/wiki/index.php/10_DOF_Sensor_(SKU:SEN0140)) or Grove IMU 10DOF (http://wiki.seeedstudio.com/Grove-IMU_10DOF_v2.0/)	Heading, Humidity, Temperature and Pressure
GPS-MTK3339 (https://learn.adafruit.com/adafruit-ultimate-gps/downloads)	Latitude and Longitude
UV sensor-SEN0175 (https://wiki.dfrobot.com/UV_Sensor_v1.0-ML8511_SKU_SEN0175)	UV index
Liquid Level-SEN0205 (https://www.dfrobot.com/wiki/index.php/Liquid_Level_Sensor-FS-IR02_SKU:_SEN0205)	Whether the Buoy is submerged or not
Dissolved Oxygen Sensor-SEN0237-A (https://wiki.dfrobot.com/Gravity__Analog_Dissolved_Oxygen_Sensor_SKU_SEN0237)	Dissolved Oxygen

**Table 2 sensors-19-04695-t002:** Parameters analysis.

Measured Parameters	Units	Aquaculture Facility	Goal
Water temperature	${}^{\circ }$C	Earthen ponds (outdoor) and RAS (recirculated aquaculture systems) (indoor)	Keeping cultured organisms within optimal thermal ranges to enhance growth and avoid mortality
Depth	meters	Earthen ponds and RAS systems	Monitor water level
pH levels	0–14 pH index	Earthen ponds and RAS	To impair the occurrence of acidosis that can be lethal to cultured organisms (namely due to the build-up of CO${}_{2}$ in the water)
Eletrical Conductivity	5–200,000 μS/cm	Earthen ponds and RAS	Secure optimal salinity to farm brackish water and marine organisms; detect abrupt shifts in salinity due to extreme weather events (e.g., heavy rainfall)
Levels of Turbidity	JTU (Jackson Turbidity Unit), 1JTU = 1 NTU = 1 mg/L (https://en.wikipedia.org/wiki/Turbidity)	Earthen ponds	Avoid the clogging of gills of filtering organisms (e.g., bivalves) and detect potentially harmful microalgal blooms
Dissolved Oxygen	0–20 mg/L	Earthen ponds and RAS	Secure that oxygen levels do not drop below critical levels and trigger a generalized mortality of cultured organisms
UV index	UV-A and UV-B intensity (mW/cm${}^{2}$)	Earthen ponds	Help decision making on the use of shading to impair damage on cultured organisms, namely on shallow systems
Humidity, Aerial temperate and Pressure	%, ${}^{\circ }$C and hPa	Earthen ponds	Help on the detection of extreme weather events (e.g., heat waves, extreme winds, heavy rainfall) that may put at risk the organisms being cultured

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
