# Peer review of "A Platform of Unmanned Surface Vehicle Swarms for Real Time Monitoring in Aquaculture Environments"

_sensors, 2019, doi:10.3390/s19214695_

Round 1

Reviewer 1 Report

"IoT Platform for Real Time Aquaculture Monitoring" presents a study where they develop a platform for collecting data in aquaculture environments using wireless communication/IoT and USVs. I think it certainly is a very interesting study, and apparently it builds upon previous developments from the same scientific group, which I consider to be a strength. The authors outline their work in a good manner, and it is quite clear how they contribute to the scientific world through this study (although they could be clearer on a couple of things, see specific comments below). The English us quite good, and the messages conveyed are quite easy to grasp, which is a plus.

However, after reading this article, I am left wondering if the title is quite descriptive of the study.

I think the use of the term IoT in the title is misleading, at least with respect to what I consider to be the most interesting part of the study, namely the use of a swarm of vehicles. Instead I suggest the authors call it something like “A system using swarms of unmanned surface vehicles for real time monitoring of marine fish farms” (my suggestion is maybe not a great title, but you get the picture). When I read “IoT” in the title, I think more about stationary nodes using radio comm to transfer data to the user. Using a swarm of vehicles is much more interesting! Although the title alludes to aquaculture, aquaculture is only mentioned in the introduction and partly in the previous work section. I think the authors need to pick up the aquaculture angle towards the end of the article to justify using aquaculture in the title. As it stands now, the last 17 pp of the manuscript treats the system largely as an aquatic platform for sampling. Now, there’s nothing wrong with treating it as an aquatic platform, but if they wish to keep aquaculture in the title, I think it important that they spend some space discussing e.g. possible applications in aquaculture, implications for aquaculture monitoring/operations etc. towards the end.

I think this is an interesting study in general (i.e. both with aquaculture and other aquatic applications in mind), and believe it will be of interest to many in the scientific world, and possibly also industry.

Based on these reflections, I recommend that the study is accepted for publication if the authors resolve the aquaculture conundrum I discuss above and respond to my suggestions below.

Specific comments:

In general: to my mind, this work seems to be relevant for the concepts of Precision Fish Farming in using technology and automation principles to improve operations in aquaculture, and make these less dependant on humans and hence more objective. I think the authors should exploit this by referring to “Føre, M., Frank, K., Norton, T., Svendsen, E., Alfredsen, J.A., Dempster, T., Eguiraun, H., Watson, W., Stahl, A., Sunde, L.M., Schellewald, C., Skøien, K.R., Alver, M.O., and Berckmans, D., 2018. Precision fish farming: A new framework to improve production in aquaculture. biosystems engineering, 173, pp.176-193.”, thus illustrating how their study is relevant also in a bigger context. Related work L 71: you could here also cite “Shi, B., Sreeram, V., Zhao, D., Duan, S. and Jiang, J., 2018. A wireless sensor network-based monitoring system for freshwater fishpond aquaculture. Biosystems engineering, 172, pp.57-66.”, as this is a study doing exactly this in pond aquaculture. L76-78: I do not understand what is referred to by “…deposited by the food” in this sentence. Does this refer to the pollution/nutrients emitted by marine fish farms? Aquatic platform L139: misspelling of “interoperability” (second “i” is missing) L155: I find the expression “prevent death” a bit unclear. Do they mean death of employees or fish? In case of the former, I would like some more information on this, as although aquaculture is a risk prone profession, I don’t think there at that many deaths. In case of the latter, I would prefer the expression “mortality” over “death”, and would link it with welfare too (i.e. something like “prevent suboptimal welfare and mortality”) L169: I would remove one “in” here, i.e. “…it can increase in the number of devices…” L171: this should be reformulated for clarity, e.g. “…mission when a vehicle malfunctions or shuts down.” L205: as far as I gather, the periodic routine referred to here involves periodic reading of sensors. You should say this explicitly here, e.g. “…and a routine where sensors are read periodically.”. I had to read the paragraph once more to understand it. No harm in making it easier for the reader. L207: reformulate, “…it reads all the sensors five times..” Figure 6: this figure is OK, but the caption should mention that this is the architecture for an USV, as there would be no point in having a path planner for a buoy? Network communication system L242: here they refer to figure 9, which indeed illustrates the point made here. I would however, prefer if figure 9 was moved to here (and being renamed figure 7), as the figures should appear in the sequence they are cited in the text (as it is now, figs 7 and 8 are between this line and fig 9). USVs’ missions L316-317: rephrase for clarity, e.g. “The group identification of the sensors to be used in the mission is also specified.” Data acquisition and analysis L350-351: rephrase for clarity, e.g. “…FIWARE, a data model adhering to the FIWARE rules had to be created.” The dashboard and mobile app are mentioned here, but it is not explicitly stated whether these two were realised as a part of the present study, or developed before this study. If you developed these here, you should say so. There are quite a few screenshots describing the dashboard application in this section. Perhaps these could be combined into one figure with sub-figures? Or perhaps some of them could be included as supplemental material? I think the text surrounding these explain how the dashboard works quite well on its own. Case study I would like this section to start with a clear statement of what this case study was set up to investigate. That way, it is easier to evaluate the results. L415-417 + figure 19: given that the most interesting stuff about this study is the data collection and communication parts, I would reduce the emphasis put on battery life for the motors here. I think a single sentence stating that the batteries were found to last for 75 mins when the motors were run continuously, and not use a figure for this. Figure 21: I don’t think this figure should be included, as 100% reception is 100% reception, and as long as this is written in the text, this is sufficient. Conclusions This could be a nice place to link the work with aquaculture again (see my first comments above for more on this).

Author Response

Dear reviewer,enclosed you can find a detailed answer to all of your questions, as well as the answer to all the remaining reviewers.  We hope our responses will satisfactorily address your comments. Regarding to the manuscript, the changes originated by the current revision are highlighted in yellow. 

Reviewer 2 Report

The paper presents the functionalities and architecture of the aquaculture monitoring system based on the cooperation between USVs, Buoys and the shore segment using the Internet of Things technology. The logic of communication between systems components has been presented in details. The implementation of communication technology types used for data transfer between system components is presented in clear and communicative way. Presented system is the example of the usage of swarm of USVs, which is the developing approach to unmanned objects utilization. Presented ideas can be used for many types of USVs control system, which implies a question to the author:

In what way the system is specific for aquaculture monitoring? Are the sensors installed on both USVs and buoys provide enough data for real vegetation conditions assessment. Does the size of AUVs allow to expand the choice of sensors installed?

Authors present the field tests of the proposed system. According to the maps presented, some of them were conducted on land. While this is acceptable approach in terms of communication testing, with floating devices the impact of sea conditions cannot be neglected.

Were authors conducted longer tests of floating devices to make sure the prototype USVs and Buys are able to maintain the functionalities while operating on open water? Or is this the area of further development?

The questions listed above  require a comment to achieve the complete understanding of the issue investigated by authors and the ideas presented in the paper.

The work is a comprehensive guide of the data transferring options available for small surface platforms, and as it is, is a valuable resource in the field of remote environmental monitoring.

Author Response

(The authors gave the same response as above.)

Reviewer 3 Report

The paper describes a proposal of a IoT-based solution for monitoring environmental and marine parameters in nearby coastal areas, with potential to be used in the aquaculture domain. The paper is well posed, it is clear and concise. The proposal is interesting but should improve in the following aspects:

The authors state that the proposed platform is oriented to the context of aquaculture. And it is certainly a system that allows collecting values ​​of different parameters related to the marine environment. However, it is not clear that the system covers the needs inherent in the aquaculture domain. In the area where I usually live there are farms for marine species that require constant monitoring of different parameters, including several that do not seem to contemplate the proposed solution, such as the direction of water movement or, above all, the presence of different toxins and contaminants. The authors should clarify the contexts in which their proposal is really applicable, within the domain of aquaculture, or at least explain the limitations and real scope of the usefulness of the proposal in that domain. The case study described in section 7 is too limited. It could be enough to verify the correct functioning of certain aspects of the proposal, but it does not show its potential utility in aquaculture productions. The description of a more realistic case study, which, although hypothetical or simulated, would contemplate the existence of artefacts of a specific aquaculture production, a particular marine extension and certain environmental conditions, would be appreciated. Based on this scenario, the potential of the proposed platform should be analyzed and described, indicating issues such as how many USVs and buoys would be necessary, what distances the vehicles would travel, how long, how often the batteries should be recharged, what problems the temporary inclemencies would entail, etc. It should also analyze what happens when some key element fails.

In the manuscript I do not appreciate significant writing errors, except that in line 197 reference is made to Figure 4a, when it should be to 2a, if I am not mistaken. The dashboard figures (12 to 16) are also strange, as they refer to drones or locations and parameter values on land, not at sea.

Author Response

(The authors gave the same response as above.)

Round 2

Reviewer 3 Report

The authors have not made significant changes in the manuscript aimed at resolving the issues identified in the first version of the paper. It is understood that the main novelty of the proposal lies in the use of a series of solutions and technologies (previously proposed and already published by the authors) in the context of aquaculture.

Obviously, it is not necessary to detail in the present manuscript the validation of these previously published solutions and technologies. But it is needed to describe the real potential of the proposal in the context of aquaculture. A holistic study of this context is impracticable in a short time. It would be sufficient to raise a realistic case study (section 7), involving a specific aquaculture production. This production/farm can be simulated or hypothetical, but it must highlight the needs and characteristics of monitoring of aquatic and atmospheric parameters in that production. Based on these needs, it would be necessary to describe how the proposed solution would be instantiated (necessary equipment, missions, operative) and should be described how it contributes respect to more traditional monitoring mechanisms. The limitations of the proposal can also be introduced in this case study (the lack of appropriate sensors for certain monitoring needs, as indicated by the authors in the reply of the previous review, should be mentioned and discussed in the manuscript. It does not detract the proposal).

I maintain my opinion about the high potential interest of the paper. But the lack of analysis of the monitoring needs in the domain and the low concretion of the realistic use of the proposal in aquaculture productions devalue the relevance of the work. Extending section 7 significantly (as discussed in the previous paragraph) would resolve this issue.

Author Response

Dear reviewer,enclosed you can find a detailed answer to all your questions.  We hope our responses will satisfactorily address your comments.Regarding to the manuscript, the changes originated by the current revision are colored in blue.
